# Perceiving depth beyond sight: Evaluating intrinsic and learned cues via a proof of concept sensory substitution method in the visually impaired and sighted

Amber Maimon[1,2☯]*, Iddo Yehoshua Wald[1,3☯], Adi Snir[1], Meshi Ben Oz[1], Amir Amedi[1]

1 Baruch Ivcher Institute for Brain, Cognition, and Technology, Reichman University, Herzliya, Israel,
2 Computational Psychiatry and Neurotechnology Lab, Ben Gurion University, Be'er Sheva, Israel, 3 Digital Media Lab, University of Bremen, Bremen, Germany

☯ These authors contributed equally to this work.
* amber.maimon@runi.ac.il

## Abstract

This study explores spatial perception of depth by employing a novel proof of concept sensory substitution algorithm. The algorithm taps into existing cognitive scaffolds such as language and cross modal correspondences by naming objects in the scene while representing their elevation and depth by manipulation of the auditory properties for each axis. While the representation of verticality utilized a previously tested correspondence with pitch, the representation of depth employed an ecologically inspired manipulation, based on the loss of gain and filtration of higher frequency sounds over distance. The study, involving 40 participants, seven of which were blind (5) or visually impaired (2), investigates the intrinsicness of an ecologically inspired mapping of auditory cues for depth by comparing it to an interchanged condition where the mappings of the two axes are swapped. All participants successfully learned to use the algorithm following a very brief period of training, with the blind and visually impaired participants showing similar levels of success for learning to use the algorithm as did their sighted counterparts. A significant difference was found at baseline between the two conditions, indicating the intuitiveness of the original ecologically inspired mapping. Despite this, participants were able to achieve similar success rates following the training in both conditions. The findings indicate that both intrinsic and learned cues come into play with respect to depth perception. Moreover, they suggest that by employing perceptual learning, novel sensory mappings can be trained in adulthood. Regarding the blind and visually impaired, the results also support the convergence view, which claims that with training, their spatial abilities can converge with those of the sighted. Finally, we discuss how the algorithm can open new avenues for accessibility technologies, virtual reality, and other practical applications.

**Data Availability Statement:** The data has been uploaded to OSF and made publicly available at the following doi: 10.17605/OSF.IO/7K26C.

**Funding:** This work was supported by a European Research Council Consolidator grant (773121 NovelExperiSense), a European Research Council POC grant (101123434 TouchingSpace360), and a European Union Horizon 2020 research and innovation program grant (101017884 GuestXR) all to AA. The research reported in this paper has also been partially supported by the German Research Foundation DFG, as part of Collaborative Research Center (Sonderforschungsbereich) 1320 "EASE - Everyday Activity Science and Engineering," University of Bremen, awarded to IYW. The funders had no role in the study design, data collection, and analysis, decision to publish, or preparation of the manuscript.

**Competing interests:** The authors have declared that no competing interests exist.

## Introduction

Vision was once considered essential for developing adequate spatial perception [1]. Specifically, depth perception within the spatial surroundings is considered a visual ability underpinned by binocular and monocular visual cues. The ability to perceive depth and its development are seemingly affected by nature and nurture [2]. Babies are known to exhibit depth perception beginning very early in life, around the time that they begin to crawl [2]. Visual perception is thought to be dominant relative to auditory perception in processing spatial events [3]. However, research on the development of depth perception involving senses other than vision is limited, and the developmental trajectory is much less clear. The extent to which our perception of depth is influenced by mechanisms acquired throughout childhood has yet to be clarified.

More generally, understanding the development and organization of perceptual abilities and their processing in the brain is evolving. Our research and that of others indicates that naturally imposed critical periods can be surpassed through perceptual learning paradigms facilitated by dedicated technologies, experiences or training. For example, it has been shown that following congenital cataract removal past the closure of critical periods, children are not able to perceive in three dimensions [4]. However, these children are susceptible to depth-related visual illusions, such as the Ponzo and Muller Lyer illusions, a very short time following their cataract-removal surgery [4, 5]. These findings and others have been suggested to be part of a broader claim for re-examining the role of sensory critical periods [6–8] as periods in early life during which exposure to sensory information from the environment is crucial for development of the senses [9, 10]. This study aimed to explore these nature vs. nurture matters while investigating both intrinsic and learned cues of depth perception.

### Cross-modal correspondences

Humans are known to have both inherent and learned associations that affect their perception. Synesthesia for example, in which a sensory experience in one modality leads to perceiving another attribute or modality simultaneously [11], is on the one hand considered to be an innate condition, but has also been explored in the context of an acquired phenomenon [12]. One form of synesthesia correlates auditory pitch with size [13], and individuals with perfect pitch have been shown to share neural correlates with auditory-related (tone-color) synesthetes [14]. Synesthetic correlations between musical pitch stimuli and spatial characteristics have also been identified, with higher notes being associated with higher locations and a diagonal orientation to the right [15].

While only a marginal percentage of the population experiences synesthesia, psychological intuitions and associations thought to "connect" between certain attributes of stimuli in different senses known as cross-modal correspondences, are widespread and experienced by the general public. Some crossmodal correspondences are thought to be innate, while others learnable. The well-known Bouba Kiki effect is an association between "sharp" sounding words such as "Kiki" with sharp-edged shapes and "round" sounding words such as Bouba with rounded shapes; this effect is thought to be based on intrinsic properties as it is consistent across ages, cultures, and levels of education [16]. On the other hand, there are also numerous crossmodal correspondences that are thought to be acquired or learned [17].

In this study which aims to explore depth perception beyond sight, our interests lie in vision and audition. Numerous such associations have been shown in the visual modality., For example, research has shown that there are certain associations between shapes and colors, specifically letters (graphemes) and colors [18]. Furthermore, research on these associations has indicated that more are acquired when one learns to read [19]. Moreover, when associations

between letters and colors are induced through reading books with consistently colored letters, it can even lead to the onset of a synesthesia-like association [20].

In audition, and more specifically in the context of spatial perception, such associations can be seen for example in the Spatial-Musical Association of Response Codes (SMARC) effect. This effect involves a multimodal contingency between the sound pitch and spatial qualities. According to the SMARC effect, higher pitches are encoded as being higher in spatial location and vice versa (irrespective of the location of the source providing the audio) [21]. This association between pitch and height is one of the most well-known crossmodal correspondences, and research has even indicated that this is due to a shared area in the brain [22, 23]. It is debated if these cross-modal correspondences are innate or acquired. The innateness of the correspondence between pitch and height has been supported by studies on preverbal infants. The infants showed a preference for an animated ball that rose and fell when presented with auditory cues in congruence with the pitch-height correspondence [24]. On the other hand, this correspondence has been suggested to be semantic and possibly even emerge following language learning [17].

## Spatial perception in the blind and visually impaired

It is commonly suggested, though debated, that people who are visually impaired or blind have compromised depth perception. Some research even indicates that visually impaired people prefer to, or tend to, rely on their residual or partial sense of vision concerning spatial perception before relying on cues from other senses that are not impaired [25]. This is a matter of ongoing deliberation concerning the ability of people who are visually impaired or blind to develop and use spatial perception in general. As they lack to a lesser or greater extent the sense of vision, the spatial abilities of the blind and visually impaired are based on their intact senses, for example auditory or tactile. As such, some hold the view that these abilities are necessarily impaired relative to those of the sighted (see Gori et al. [26] for more information), and others hold the opposite view, indicating better abilities (in auditory, tactile, and higher level domains such as memory and language) owing to compensatory mechanisms [27–30]. Either way, it has been shown that training can be effective for enabling and improving spatial abilities in blind and visually impaired individuals [31, 32]. This is further supported by research into blind and visually impaired individuals who use echolocation to acquire spatial representations of their surroundings [33–35].

## Representations of spatial information via sensory substitution

Sensory substitution is the conveying of sensory information usually acquired by one sense through another sense [36, 37]. This allows for the new information to be processed in the brain akin to the substituted sense. Sensory substitution has long been employed for studying questions related to sensory perception in general and more specifically, to spatial perception. Alongside this, sensory substitution devices (SSDs) and algorithms have also been explored as potential tools for aiding visually impaired individuals [38–43].

Two central approaches have commonly been applied for representing spatial information via audio or touch. The first involves a conversion of full images to audio or tactile stimulation [42, 44–47]. On the practical level, the main advantage of such methods is their ability to preserve and convey a large amount of information present in the visual scene. This way, following a learning process, the brain is able to utilize its inherent abilities or develop new ones for comprehending various dimensions of information from the scene [48–53]. The main disadvantage of these methods relates directly to their advantage—in order to reach a meaningful level of comprehension pertaining to the scene in its entirety, extremely lengthy training times

are required. As these methods often involve a more or less arbitrary translation from one sensory modality to another, the users are in essence learning a new language, to be able to process and perceive the information being conveyed. The second approach involves simplifying the depth or distance dimension specifically to a single or several pixel resolution. The pros of this approach are the ease and straightforwardness of perceiving depth using these systems, which commonly require no training time at all [54–56]. The cons are the very low or relatively low resolution provided, often incapable of or not accounting for information in the scene other than the single dimension of distance.

In this study, we suggest a third novel, cross-modal, ecological, sensory substitution approach. This approach strives to balance between the previous two and rather than teaching a new language altogether for conveying an abundance of information, relies on the existing knowledge of language, while using sensory substitution only for representing the facet of spatial location [25, 57]. A previous algorithm which this work builds upon has been shown in prior research to be effective and intuitive for perceptual learning due to its tapping into mechanisms humans are particularly attuned to, employing a combination of language and pitch manipulations for vertical location, and time for horizontal. Thus far, the algorithm has been employed with sighted, visually impaired, and blind participants and has proven to be easily trained on and effective for conveying spatial information in both the front and backward spaces through sensory substitution on the horizontal and vertical axes [25, 57]. While the previous algorithm represented objects and their horizontal and vertical locations, this work expands the implementation by adding a representation of depth.

We propose representing depth using an ecologically inspired representation, through auditory properties based on the way auditory cues of depth are perceived naturally, hence building again on a familiar "language", rather than using more arbitrary mappings sometimes used in sensory substitution. This is in line with and supported by prior research demonstrating that using cross modal correspondences can be beneficial for enhancing the performance of sensory substitution device users [58, 59]. For a discussion on intuitive mappings explored previously in sensory substitution research see Stiles & Shimojo [60]. To serve our study's purpose and focus, we removed the representation of the horizontal axis for the sake of specificity. Two versions of the new algorithm were created to discern intrinsic correspondences and determine whether they facilitate effective learning of spatial cues related to depth through audition. 1) Depth was represented using the ecologically inspired approach proposed above, specifically based on a simulation of how sound is filtered in nature when traveling over distance, losing gain and higher frequency sounds; While vertical location was represented using pitch change as presented by the previous algorithm. and 2) an interchanged version of this representation with a higher pitch representing further objects, and the more synthesized auditory cue representing higher objects.

Our first hypothesis in this study was that an SSD that employs language and relies on known cross-modal correspondences alongside ecologically inspired cues would be more effective than one that uses arbitrary correspondences. Alternatively, this could be inconsistent among individuals, with different people finding one mapping or the other more intuitive, or finding them equally intuitive, in which case there would be no difference in performance using the system. In addition, based on the learnability of some cross-modal correspondences, we hypothesized that participants will be able to successfully use the system for spatial localization of depth information following training to a greater extent than chance. As such, this study ultimately had a number of overarching goals, the first, to translate information about depth through a repurposing of the algorithm we employed in prior research for conveying spatial information through language and auditory properties [25, 57]. Next, this study weighed in on questions concerning the inherent vs. learned nature of depth perception by

investigating whether and which mapping the participants would be able to successfully use for spatial localization in the depth dimension. Finally, we aimed to provide insight into the abilities of the blind and visually impaired with respect to this form of auditory depth perception, and compare their training and use of the algorithm to those of the sighted.

## Methods

### Participants

40 individuals over the age of 18 participated in the study (13 males and 27 females, *Mean age* = 29.2 years old, *SD* = ±10.8). Out of the participants, five were blind (2 males and 3 females, *Mean age* = 41.8 years old, *SD* = ±12.6), and two were visually impaired (2 males, aged 55 and 26). Blind and visually impaired participants in the study were determined via a certificate of blindness. A person is entitled to receive the certificate if they are totally legally blind or with visual acuity of 3/60 meters [61]. All non-blind participants were required to wear a blindfold throughout the experiment. All participants reported normal hearing and neurological function. The Reichman University Department of Psychology IRB committee approved the study (Ethical approval P_re1_2022036), and all participants signed written informed consent forms. Recruitment for the study took place between May and December 2023 through the Reichman University SONA participant recruitment system, social media, and through personal connections to the visually impaired community. With respect to the experiment conditions, there were 20 participants in each condition: 17 sighted, 2 blind and 1 visually impaired in one condition (Original); and 16 sighted, 3 blind and 1 visually impaired in the other condition (Interchanged). All participants (sighted, blind, and visually impaired) were randomly allocated to one of the two conditions.

### The TopoLanguageDepth (TLD) algorithm

The algorithm aims to represent a scene's content using language by an auditory cue naming the objects (e.g bottle, spoon), while indicating the object's location using auditory features applied to the cue. For focusing this study, we represented only the depth (z-axis) and elevation (y-axis) of the space.

The z-axis location was represented using a modulation based on the natural behavior of sound in distance—two features were used for this purpose 1) to simulate how higher amplitude is perceived to be closer and vice versa corresponding to a well-known cross-modal correspondence, the amplitude of the sound was manipulated to be inversely proportional to the distance [62]. 2) To emulate the quicker absorption of high frequencies as opposed to lower frequencies in nature, subtractive synthesis can be used to simulate the change in sound as it naturally makes its way through the air. For this purpose, a low-pass filter (that removes higher frequencies gradually) was used [63]. The parameters used for Close, Middle and Far were (respectively) Cutoff Frequency (CO): 24 kHz, Quality Factor (Q): 1.0, and Gain (G): 0.13; CO: 5 kHz, Q: 1.0, G: 0.13; and CO: 0.8 kHz, Q: 0.7, G: 0.3. It is important to note these parameters were chosen to represent a noticeable difference in distance by sound, as caused by physical effects such as filtering due to propagation through the atmosphere. These changes are often more noticeable in greater distances, and so this synthesis aims to convey distance in an effective manner inspired by ecological effects, rather than simulate the exact variation in sound dependent on its associated location.

The y-axis location was represented by pitch, as in the previous algorithm, based on the well-known cross-modal correspondence between pitch and elevation, as mentioned above. The parameters used were: Low C, Low A#, and Middle G#, with higher pitch representing higher vertical location, and vice versa. For more information on the previous algorithm, see

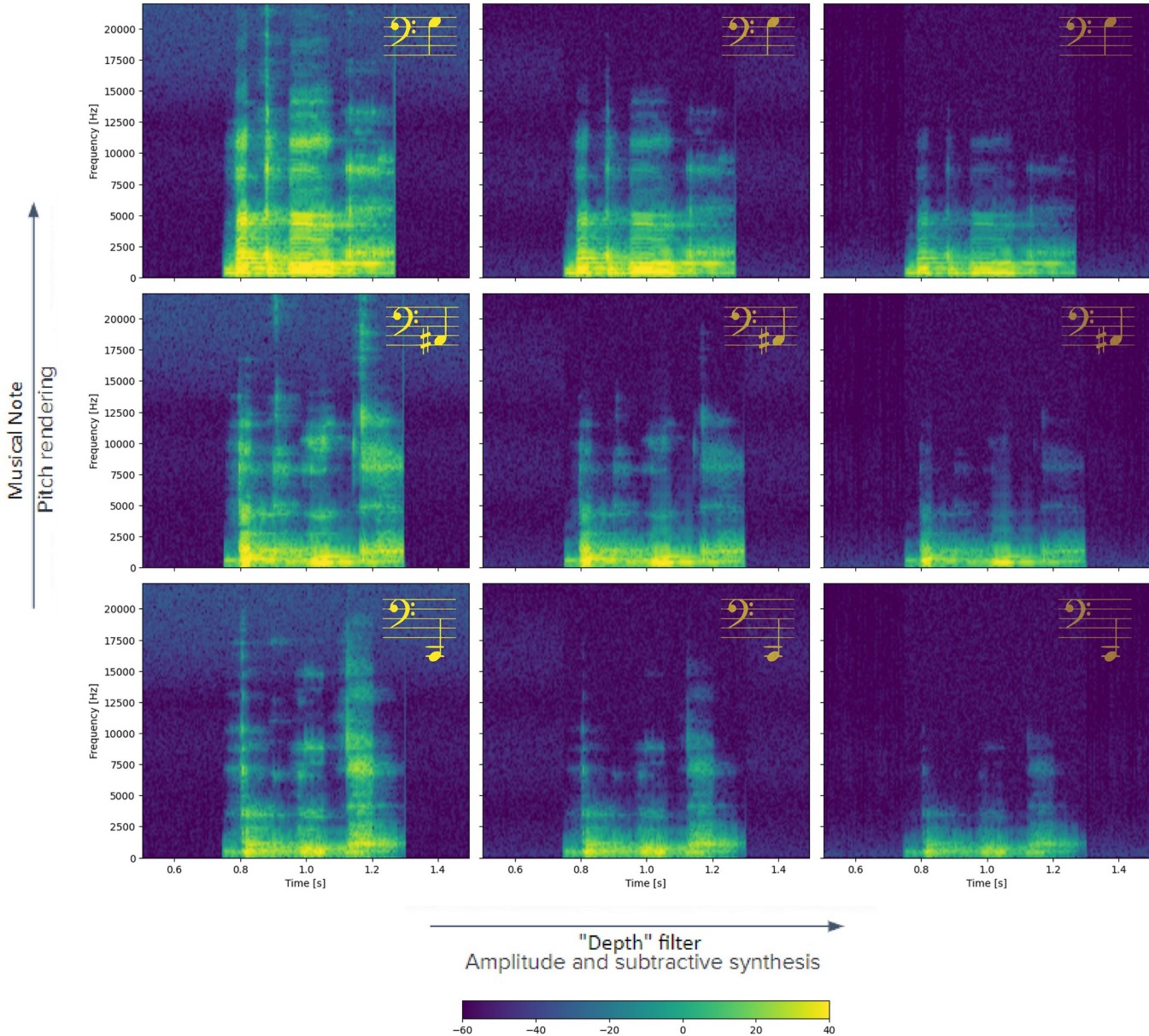

**Fig 1. Spectrograms of one of the audio cues used in the experiment with its different variations based on location.** The word played was "Bakbuk", meaning bottle.

[25, 57]. The effects of applying the different features on an auditory cue can be seen in an example in Fig 1, and how these are applied in the two different conditions is Fig 2.

## Experimental design

A between-participants, two condition design was used—Original and Interchanged. In the original condition, the TLD algorithm was used, with the y and z mapped based on nature based considerations and cross-modal correspondences as described above. In the interchanged condition the representation of the two axes was interchanged, with a higher pitch representing further objects, and the more synthesized auditory cue representing higher

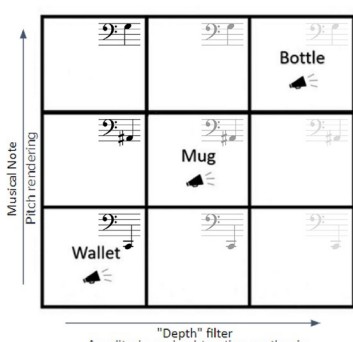
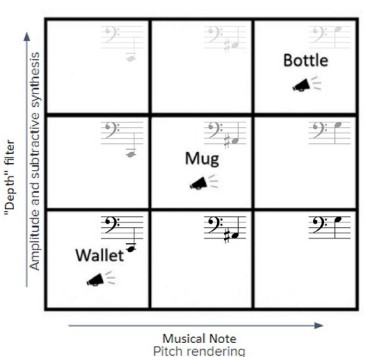
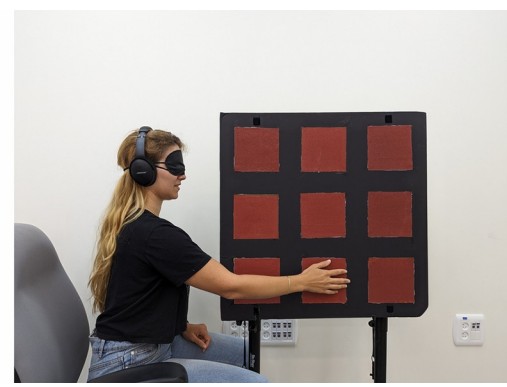

**Fig 2.** Illustration of the algorithm in both conditions—Original (left) and Interchanged (center), and its implementation in the test environment (right).

objects. The experiment consisted of 3 stages—Baseline, Training, and Test—in which participants were presented with auditory cues, and asked to localize the presented objects on a 3x3 location grid.

Before entering the experiment room, the participants received a general explanation of the experiment, after which they were blindfolded and led to the room. Upon entering the room, participants were seated perpendicularly to a 3x3 grid, so their shoulders aligned with the middle row (see Fig 2). The grid consisted of an 80x80cm board mounted on two poles. Nine 10x10cm square pieces of sandpaper were glued on the board at equal distances. Sandpaper was used to allow for easy tactile recognition of the different cells. Participants received a brief explanation about the grid, that it is composed of three rows and three columns, and that the textures represent the cells, and were told to reach out and touch each of the cells.

During each phase, participants were asked to indicate the location correlated with the presented cues in two ways: First, participants had to place their hand on the chosen cell and, at the same time, verbally announce in which location their hand was. The words "Up," "Middle," and "Bottom" represented the y-axis, and "Close," "Middle," and "Far" represented the z-axis (e.g., the position "Up Close" refers to the cell located in the top row and the closest column to the participant). If the participants' hands and spoken responses did not correspond, they were asked to clarify their answers.

At the baseline stage, the participants did not receive any explanations as to the mechanism of the algorithm but were rather just told to listen carefully to the stimuli and try to locate it on the board according to their impression. Each word was presented twice before participants were asked to indicate the represented location. This stage consisted of 15 trials.

Before starting the training stage, the mechanism of the algorithm respective to the experimental condition was presented to the participants. Throughout the training, participants received verbal feedback on their responses, whether they were correct or incorrect. If they were correct, the stimulus was played once more before continuing to the next trial. Otherwise, the cue was played again, and participants were asked to try localizing it again. In case a participant made three consecutive mistakes, the experimenter would direct them to the correct row or column, in relation to the location indicated by the participant in their last answers. If after direction the chosen location was yet again incorrect, the correct answer was revealed. The training consisted of 27 trials, and included nine named objects, that appeared at three different locations on the board.

Before starting the test phase, a 2-minute break was given, during which participants remained blindfolded. At the test stage, same as the baseline, participants were presented twice

with each auditory cue, and were asked to indicate the location correlated with each presented cue, based on their training in the previous stage. Participants did not receive any feedback on their responses. This stage contained 90 trials, and included ten named objects, each of which appeared at each one of the nine locations on the board.

Finally, participants underwent a short semi-structured interview outside of the experiment room based on nine open questions, relating to their experience, challenges, and general queries regarding their perception of depth in daily life. The aim of the interview was to gain subjective qualitative insights in addition to the quantitative results.

## Statistical analysis

Numerous statistical tests were utilized to analyze the data. A paired sample t-test was employed to compare the means of related groups, specifically to examine the differences in success rates between the baseline and test stages within the same group for both conditions (Original and Interchanged). This test was also used to compare success rates separately for the y-axis and z-axis within each condition. Independent sample t-tests were used to compare the means of two independent groups: to compare the success rates between the original and interchanged conditions in the baseline stage, as well as to compare success rates in the test stage between these two conditions. Furthermore, independent sample t-tests were used to assess the differences in success rates between sighted and blind and visually impaired participants in both Baseline and Test stages. Independent sample t-tests were also utilized to compare training durations between the original and interchanged conditions and between sighted and blind and visually impaired participants.

## Results

### Participants in both conditions showed significant differences between their baseline and test success rates

A paired sample t-test showed a significant difference between baseline and test success rates in the two conditions: Original $t(19)$ = -12.94, $p < .001$, *Cohen's d* = -2.36, interchanged $t(19)$ = -10.56, $p < .001$, *Cohen's d* = -2.89 (see Table 1 and Fig 3). Fig 4 shows the distribution of the baseline and test results. This significant difference also holds for each axis separately: y-axis: significant difference between success at baseline vs test in the original condition $t(19)$ = -6.05, $p < .001$, Cohen's $d$ = -1.35 and in the interchanged condition $t(19)$ = -11.02, $p < .001$, Cohen's $d$ = -2.46, and z-axis: significant difference between success at baseline vs test in Original condition: $t(19)$ = -6.25, $p < .001$, Cohen's $d$ = -1.4, and Interchanged condition: $t(19)$ = -8.14, $p < .001$, Cohen's $d$ = -1.82.

### Participants in both conditions achieved similar success rates in the test stage following brief training, indicating the learnability of both mappings

In the test stage, an independent sample t-test showed no significant difference between success rates in the two conditions (Original and Interchanged) ($p$ = .907) (See Table 1 and Fig 5).

**Table 1. Mean success rates of the participants in the baseline and test stages.**

|  | Baseline Mean (SD) [CI] | Test Mean (SD) [CI] |
|---|---|---|
| **Original condition** | 20.35% (±10.95) [15.22, 25.48] | 59.5% (±10.65) [54.52, 64.48] |
| **Interchanged condition** | 10.05% (±9.73) [5.49, 14.61] | 58.95%,(±17.99) [50.54, 67.36] |

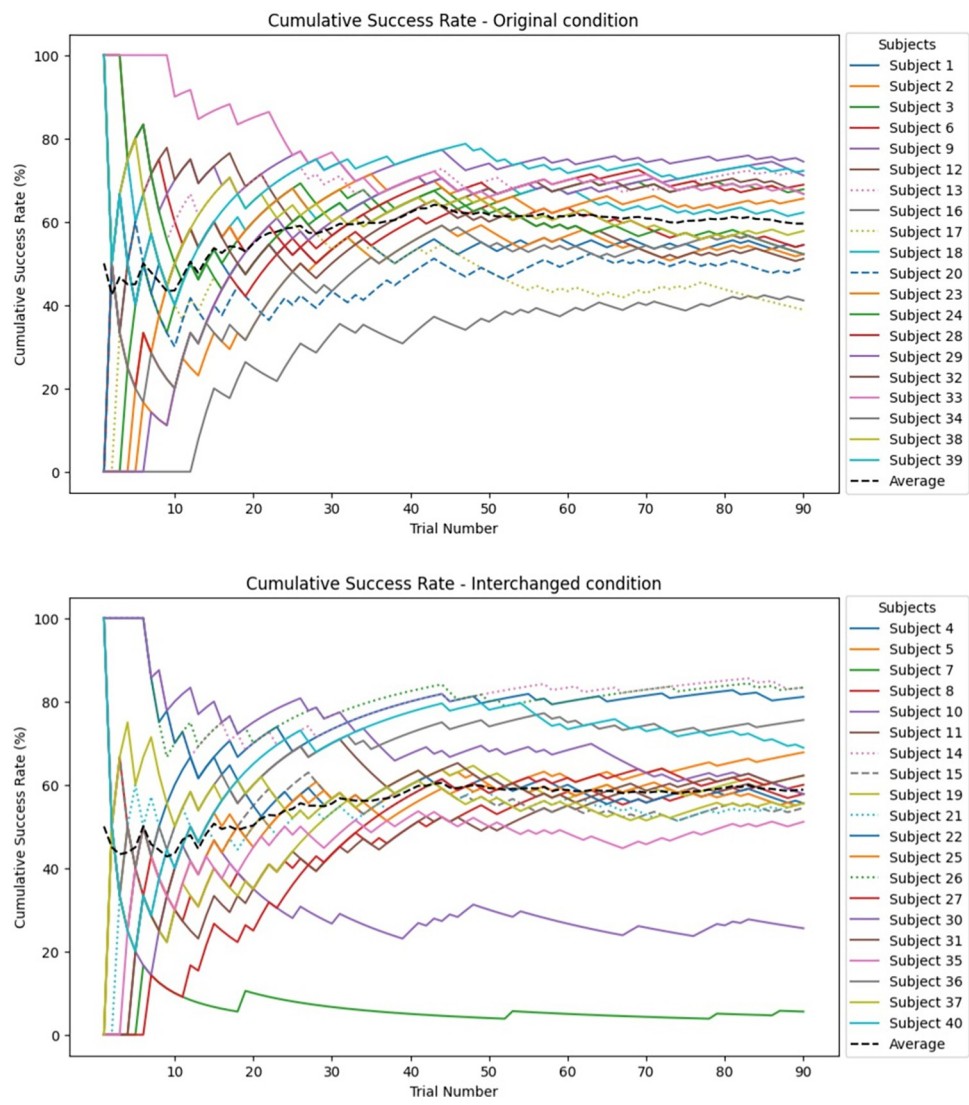

**Fig 3. Cumulative success rates of the individual participants by trial.** Dotted lines represent participants who are blind, dashed lines represent visually impaired.

## Participants in the original algorithm group had significantly higher success rates than participants in the interchanged group at baseline

In the baseline stage, an independent sample t-test showed a significant difference between success rates in the two conditions (Original and Interchanged): $t(38) = -3.14$, $p = .0015$, *Cohen's d* = -.994. The participants in the interchanged group scored around the chance level of 11% at baseline. This result indicates the intuitiveness of the mappings reliant on inherent correspondences compared to the mapping which required learning.

In the baseline stage, an independent sample t-test showed no significant difference between success rates between the sighted and blind over the two conditions (Original and Interchanged) ($p = .322$). When divided into the different conditions, there were no significant statistical differences between the sighted and blind in the baseline stage in the original condition ($p = .066$) and interchanged condition ($p = .668$). Similarly, in the test stage, an

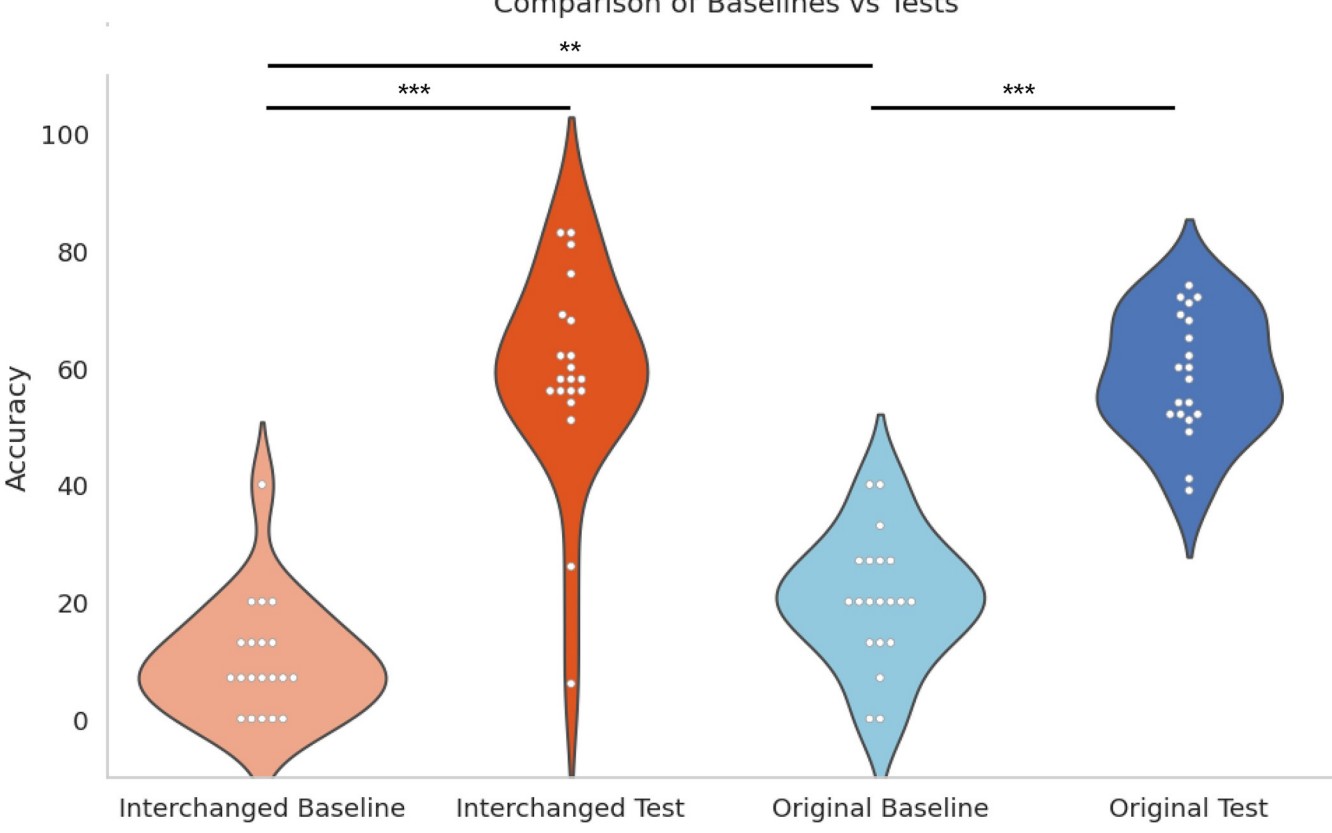

**Fig 4. Comparisons between baseline and test in both conditions.**

independent sample t-test showed no significant difference in success rates between the sighted and blind over the two conditions (Original and Interchanged) ($p$ = .548). When divided into the different conditions, there were no significant statistical differences between the sighted and blind in the test stage (Original—$p$ = .289, Interchanged- $p$ = .221). Due to the lack of significant differences between the success rates of the sighted and blind/visually impaired individuals, the results were pooled for the analysis of the success rates. This also indicates that the abilities of the blind and visually impaired converged with those of the sighted following the training in these algorithms.

### Participants in both conditions showed similar training durations

An independent sample t-test showed no significant difference between the training times of the original and interchanged conditions over the different groups ($p$ = .670) (see Table 2).

### Success rates were differentially distributed across the different locations

The highest average success was recorded in both conditions on the high-pitch, least-synthesized cues, while the lowest average success rate was recorded on the low-pitch, least-synthesized cues (see Fig 6 showing success rate by cell. The "flipped" representation is presented to enable intuitively comparing between the performance in response for each sound in the different conditions).

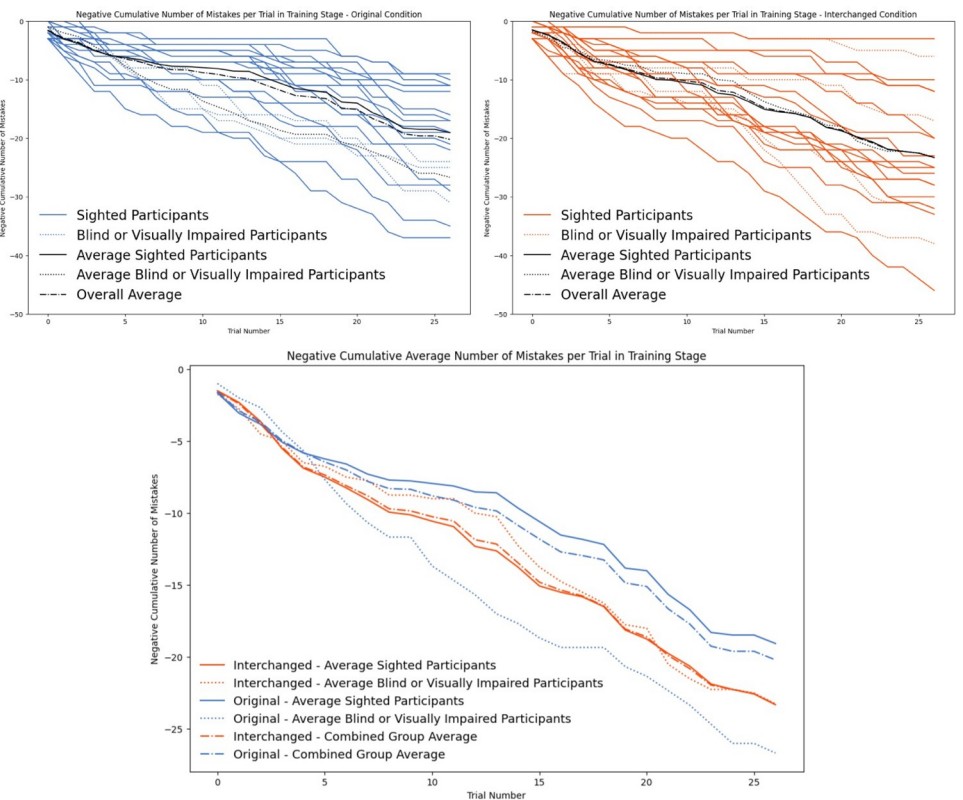

**Fig 5. Negative cumulative number of mistakes per trial in the training stage.**

### In the Interchanged condition, during the test stage, participants made significantly fewer mistakes in estimating depth (pitch modulation) than in elevation (subtractive synthesis)

Using paired sample t-test, in each condition separately, while no significance was found in the original condition ($p = .349$), a significant statistical difference was revealed between y and z-axis in the interchanged condition: $t(19) = -5.10$, $p < .001$, Cohen's $d = -1.141$. No significant differences were found in either condition when comparing the y and z axes of the baseline stage (Original—$p = .856$, Interchanged—$p = .120$). (See Fig 7 showing violin plots of the success rates by axis, Table 3 showing success rates per axis, and Fig 8 showing heatmaps of the success rates per cell location).

While most results did not show a significant difference between the blind and visually impaired individuals compared to sighted performance, *the blind and visually impaired participants took significantly longer to train than the sighted participants*. Table 2 shows the average training durations divided by groups and conditions. An independent sample t-test revealed statistically significant differences between sighted to blind individuals in training times: $t(36)$

**Table 2. Average training durations for both conditions.**

|  | Original condition | Interchanged condition | Overall |
|---|---|---|---|
| **All participants** | 00:14:15 (±00:09:32) | 00:15:38 (±00:10:27) | 00:14:57 (±00:09:54) |
| **Sighted** | 00:13:09 (±00:09:05) | 00:12:44 (±00:06:51) | 00:12:56 (±00:08:26) |
| **Blind/visually impaired** | 00:20:12 (±00:04:17) | 00:26:24 (±00:15:22) | 00:23:50 (±00:11:39) |

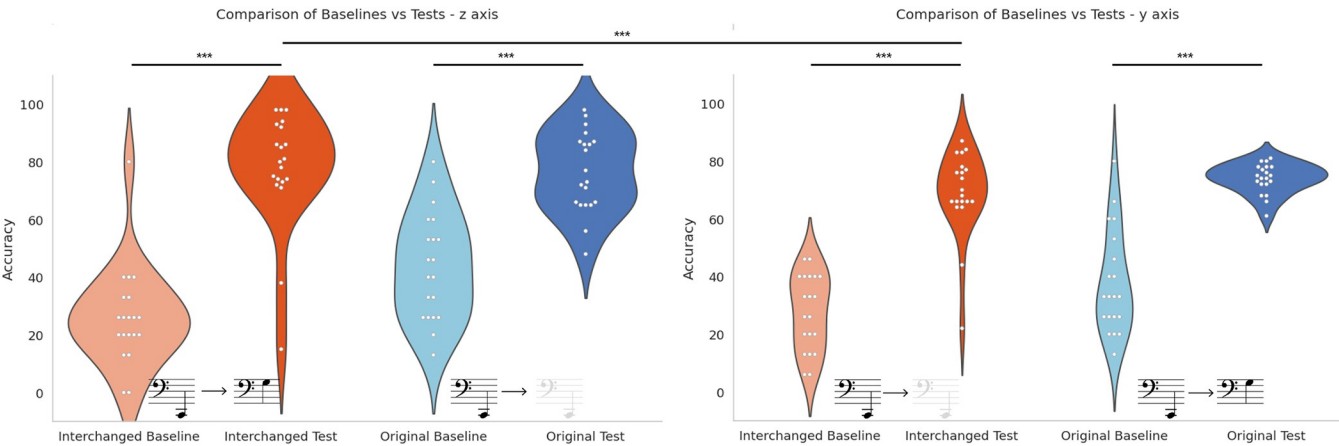

**Fig 6. Distribution of success rates in the different cells per condition.** Original condition (top left), Interchanged condition (top right), Flipped diagram of the interchanged condition (bottom left). The flipped diagram represents the same data, with the axis flipped, so that the grid matches the auditory representation of the original condition.

**Fig 7.** Success rates of baselines and tests for each condition divided by axis—z axis (left) and y axis (right).

**Table 3. Average success rates per condition, per axis.**

| | Z-axis | | Y-axis | |
|---|---|---|---|---|
| | **Baseline Mean (SD) [CI]** | **Test Mean (SD) [CI]** | **Baseline Mean (SD) [CI]** | **Test Mean (SD) [CI]** |
| **Original condition** | 44.67 (±19.84) [35.38, 53.96] | 74.83 (±4.84) [72.56, 77.10] | 46.01 (±20.85) [36.25, 55.77] | 77.67 (±14.02) [71.11, 84.23] |
| **Interchanged condition** | 37.33 (±21.78) [27.14, 47.52] | 80.50 (±16.99) [72.55, 88.45] | 27 (±12.88) [20.97, 33.03] | 69.61 (±15.13) [62.53, 76.69] |

= 2.87, $p$ = .007, *Cohen's d* = 1.203. Due to the low number of blind and visually impaired participants, this should serve as an initial exploration and cannot be generalized (see also Fig 5).

## Discussion

This work presents the TLD sensory substitution algorithm, which translates information about content, spatial depth, and elevation into a combination of language and auditory features, designed based on known cross-modal correspondences and ecological principles. The algorithm was employed for exploring representations of depth and aspects of perceptual learning. The experiment showed that 1) Participants in both conditions demonstrated significant differences between their baseline and test success rates. 2) Participants in both conditions

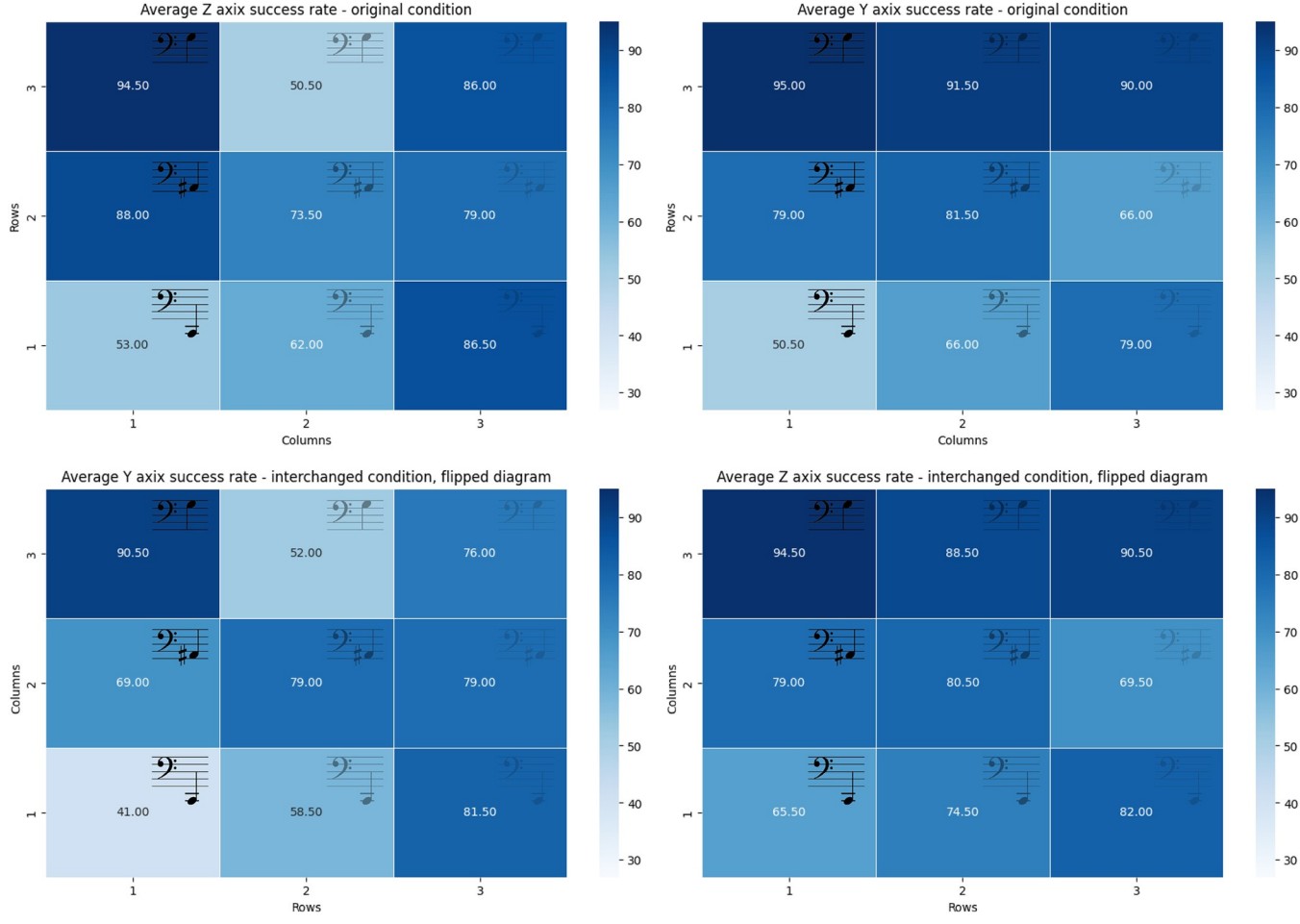

**Fig 8.** Heatmaps of the success rate per cell location per axes in the original condition (left) and the interchanged condition (flipped diagram) (right). The left and right grids are aligned to match in their auditory representation.

achieved similar success rates in the test stage following brief training. 3) Participants in the original algorithm group had significantly higher success rates than participants in the interchanged group at baseline. 4) Participants in both conditions showed similar training durations. 5) Success rates were differentially distributed across the different locations. 6) In the interchanged condition, during the test stage, participants made significantly fewer mistakes in estimating depth (pitch modulation) than in elevation (subtractive synthesis). 7) The blind and visually impaired participants took significantly longer to train than the sighted participants.

Following our hypothesis, the higher success rate of participants in the original condition compared to the interchanged condition at baseline, indicated the intuitiveness of the correspondence between the chosen acoustic characteristics. Also, according to the hypotheses, and in line with prior research on this algorithm and others for conveying visual information through audition, people were consistently significantly able to improve their ability to perceive spatial location using the algorithm following a very brief period of training. Surprisingly, there was no significant difference between participants' ability to train on depth perception in the original and interchanged conditions, indicating a high effectiveness for the training, and a high general learnability for these mappings. Initial results on the blind and visually impaired participants did not indicate a significant difference from the sighted in their ability to successfully learn and use the algorithm, though they did take longer to complete the training. This is in line with the claim that while the brain is most plastic at a young age, new sensory capabilities can be trained past the closure of commonly accepted critical periods [6].

## Insights from the system functionality

Sensory substitution methods are utilized both for basic science—gaining a deeper understanding of our sensory perception in behavior and in the brain, and for practical purposes—such as rehabilitation, sensory enhancement, and more. As such, there is an ever-growing desire to elucidate a variety of sensory attributes by employing these methods. Depth perception is a central aspect of the vital cognitive and functional skill of spatial perception and is of great interest in the realms of sensory substitution.

## Previous depth representation SSDs

Several methods have been explored in the past for representing spatial depth information via sensory substitution. A few somewhat different yet related methods for representing images via sound rely on converting 2D-pixel information into soundscapes. Such systems allow for perceiving depth based on one's understanding of perspective, relying on "the perceived size, the height in the image and the linear perspective . . . as depth cues to estimate the distance of the targets . . . and to perceive depth" [50]. One of these systems, the Prosthesis for Substitution of Vision with Audition (PSVA) [50, 64], was used to evaluate depth perception. It was also used in a study exploring how the brain functions during this form of depth perception, indicating visual cortex activation, which is in line with previous research showing visual cortex activation in blind individuals when using similar SSDs for discerning visual features [44, 48–52]. A system based on an algorithm named MeloSee [65] used a similar concept of translating pixel information to sound, specifically conveying distance with pitch for representing height, gain manipulations for representation of left and right, and loudness for representation of depth. In a later study, the effects of modifying this system by changing these mappings for different sounds for height and adding pitch features to depth were evaluated, showing improvements in some scenarios [66]. A recent study explored five different auditory parameters (frequency, amplitude, reverberation, repetition rate, and signal-to-noise ratio between two

sounds) with respect to their effectiveness in conveying only the depth of a single location (rather than multiple pixels) and found that the repetition rate of beeps was the most effective, with close being represented by fast beeps and the opposite [67]. It is important to note that such approaches are extremely difficult to master and even after extensive training, the systems are difficult to use and manage in real time (real-life) complex environments [42, 68]. To a large extent these approaches fail to be practical enough to make the systems relevant for wide-spread use due to such limitations.

Other directions use visual-to-tactile sensory substitution for conveying depth. The Eye-Cane is a single-point (pixel) indicator of depth, indicating the distance of objects from the holder of the device with direct auditory feedback or haptic vibration represented by intensity (more intense is closer) [54, 55]. While the EyeCane system is markedly easy to master and use–a key limitation of it is that one needs to explore the environment point-by-point (pixel-by-pixel). As a result, complex environments are hard to understand and users cannot identify objects, but rather only their distance from them. Another system that provides auditory and tactile information is the Sound of Vision project, which integrates computer vision [69]. The Tactile-Sight system uses a high resolution of tactile actuators for conveying distance information through vibration on the body [70], while the Electro-Neural Vision System uses Transcutaneous Electro-Neural Stimulation (TENS) electrical impulses on the fingers [71].

The TLD system mainly differs from these systems in its simultaneous use of language for identification, alongside simple sound manipulation for conveying the location of an object while simultaneously communicating dimensions of spatial information about a scene.

## The TLD implementation of depth representation SSD

The results of the present study support those of previous studies utilizing the algorithm we presented in previous research [25, 57], indicating a high degree of learnability with training times averaging under 15 minutes overall, leading to significant improvement from the baseline and well above the 11% chance level using the system. At baseline, the original mapping significantly outperformed the interchanged mapping, indicating the utility of crossmodal correspondences and ecological cues. This implies how intuitive sensory substitution representations can be designed by relying on these principles. More generally speaking, these findings indicate systems that rely on perceptual learning combined with symbolic representations such as language, could be practical for real-life applications by being intuitive and allowing for quick proficiency with relatively low effort while conveying rich information. This stands out in comparison to some SSDs, as mentioned, that face significant drawbacks in that they require extensive training for the use of the systems and compliance with extended, often difficult training paradigms.

Looking at the heat maps conveying the distribution of success rates by location on the grid (see Fig 6), it is very noticeable that the success rates by location in the original condition closely resemble the flipped heat map of the interchanged condition. This implies that the ability to correctly recognize the locations was determined mainly by one's ability to process the signal. That is, mistakes resulted mostly from the signal perception limitations rather than the ability to perceive the correlation to the axis. This emphasizes the importance of refining the stimuli chosen. For example, one can learn from the success rate distribution which cue differentiations or combinations were more difficult for participants, and accordingly accentuate the signal manipulation as required. An interesting insight in this context arising from the distribution by axis maps (see Fig 8) is that the lower volume of the cues at the end of the subtractive synthesis spectrum (i.e. the furthest items in the original condition) did not affect participants' ability to differentiate the pitch of the cue. Also, the two heat maps representing

pitch recognition (see Fig 8, bottom figures) show that the highest pitch was the most recognizable auditory feature.

With that said, it is interesting to note that as shown in the analyses and is also apparent in the heatmaps (see Fig 8, right images), participants in the test stage of the interchanged condition made significantly fewer mistakes in the estimation of depth (pitch modulation) than in elevation (subtractive synthesis). This may imply that the less-intuitive mapping of the subtractive synthesis to height may have still affected participants' perception in the interchanged condition after the training, even if less significant than the ability to perceive the signal.

## Insights regarding cross-modal correspondences

The present study's findings indicate that while there are intrinsic, intuitive associations in sensory perception (as shown by the higher initial success in the original condition), the ability to learn and adapt to new sensory associations is also significant. Perceptual learning can drive changes in sensory systems behaviorally and neurally [72, 73] and has been extensively explored in vision, but also in audition [72, 73]. As mentioned, this study corroborates our hypothesis that there is indeed an intrinsic association with respect to depth perception for the original condition as opposed to the interchanged condition as a significant difference was shown in the baseline between these two conditions.

On the other hand, both conditions in the present study showed a similar level of rapid learnability, with a nonsignificant difference in success between conditions at the test stage, and a significant difference in participants of both conditions between the success in the test stage as compared to the baseline. This indicates that despite the inherent difference, the effect of learning is no less profound on the present association between auditory properties and depth. These findings further fit into a broader research scope, indicating 1) that both intrinsic and learned facets influence our sensory perception, and 2) that multisensory integration protocols can be particularly powerful for inducing perceptual learning [73].

Such research on multisensory perceptual learning has sparked a paradigm shift in neuroscience pertaining to how the organization and development of the brain are viewed. These insights have led to influential theories that correlate the perceptual mechanisms in the brain to metamodal [74], supramodal [38, 75], or task-specific sensory independent processing systems [40, 76]. Furthermore, it has been suggested and demonstrated that the learning effects could be more significant when the multisensory stimuli being trained on are congruent, defined as being in alignment with the individual's previous experiences or the natural relationships between the senses [73]. Training on multisensory congruent stimuli may be able to bring about novel capabilities, and may possibly induce novel mappings in the brain [77].

When asked during the semi-structured interview to report on the intuitiveness of the algorithm they were presented with, nine participants in the original condition reported unequivocally that it was intuitive (only replies that indicated a clear yes response to the question of "was it intuitive" were considered), while only two replied that the algorithm was intuitive in the interchanged condition. Among the participant reports in the original condition, it is interesting to note that participants indicated that the algorithm "Made sense, at first before I knew the rules", they indicated that "It's very easy once you understand the point". Some gave more specific information, for example, this is a reply given by a blind participant: "I think it was very intuitive, it makes sense that a weak volume would be far away, a high tone would be up".

Participants reported clear discrepancies and confusion regarding their perceptual experience in the interchanged condition. For example, one participant reported that it was: "Confusing . . . The whole first part I thought was backward. . . instinctively, it didn't feel natural to

me." Participants were very open with detailing the discrepancy between what they would have expected and what they experienced, for example: "In the beginning I actually thought it was more logical the opposite way, that the higher the sound it would be higher and if it is more powerful it would be far away. But then I got used to it and it became more connected to me." and "It was the opposite of what I thought. I thought the high would be closer to me and the low would be further away, and it was strong at the top and weak at the bottom and it was the opposite.", "It doesn't necessarily work out. It made more sense to me that the top would be stronger and not the other way around, and it was also confusing at first. With the pitch, it did make sense that what is further away is more shrill.", "It confused me a little. It didn't work out for me". It is interesting to note that two blind participants were the only two to report that they perceived the interchanged condition to be intuitive. As also detailed in the following section, these subjective reports are particularly interesting in light of the similar level of proficiency they reached in both conditions.

The findings of this study can be said to support the growing body of work challenging the classic understanding of the theory of critical periods, showing that novel sensory-perceptual mappings can be trained for well into adulthood [6, 78–80]. The findings also correspond to current theories unifying cross-modal plasticity and skill acquisition [81], and the idea that these rapid learning mechanisms can be influenced by preexisting "blueprints" or scaffolds in the existing structure of the brain [82, 83].

## Insights from the initial findings regarding blind and visually impaired individuals

Specifically with respect to blind and visually impaired individuals, it has been claimed that their spatial abilities resemble those of the sighted when functioning in the dark or with their eyes closed (i.e. when they are visually deprived) [1]. Furthermore, it has been shown that people's ability for auditory localization is reduced in the dark [1, 84]. Overall, it has been suggested that blind individuals have impaired spatial capabilities, among them spatial localization and sense of space in comparison to the sighted [26, 85, 86]. It has also been indicated that they have difficulties adequately constructing an inference of a third dimension, as this is something the sighted do based on the 2D images projected on their retinas [50]. The visual map theory supposes that blind individuals may face challenges or be slower to develop spatial skills, as vision serves an organizing role in spatial perception [1]. More specifically, it has been suggested that blind individuals have a harder time fixing positions identified through audition on a map of visual space, impacting their ultimate abilities for localization [1, 84].

This study found no differences in the abilities of blind and visually impaired individuals and the sighted with respect to their success in learning the algorithm. The present study's findings first and foremost support our previous findings using this system. They also support the greater view of convergence with respect to spatial abilities in blind and visually impaired individuals. This view posits that the spatial abilities of the sighted and the visually impaired can converge with those of the sighted (with respect to performance in spatial tasks) following training or experience; as opposed to the cumulative or persistent views that suggest that blind individuals are necessarily posed with an insurmountable disadvantage [39, 87–89].

It should be noted that in the present study, blind and visually impaired individuals took significantly longer to train on the algorithm. This may indicate that some aspects of depth perception indeed are initially limited. When broken down further, the difference in training time in the original condition in blind and visually impaired individuals was insignificant relative to that of the sighted (p = .251), while the difference in the interchanged condition specifically was significant t(17) = 2.74, p = .014, Cohen's d = 1.54.

It is worthwhile noting that the only two participants who found the interchanged condition intuitive were blind. In the semi-structured interviews, one participant explained as follows: "I think it makes sense. I understand the progression between the stages. Once I understood the meaning of the intensity it was easier for me. The tone was intuitive for me, but at the same time, I could intuitively invent another rule system for myself. It is a rule system that must be acquired because I don't think intuitively it will be similar for everyone, but it is not contrary to our basic insight." It would seem that as this participant described it they view this form of perception as somewhat fluid, as something that can be taught depending on prescribed principles. Correspondingly, it is curious to note that only blind participants reached 100 percent accuracy in any axis, specifically two reached 100 percent accuracy in the z-axis following training on the interchanged condition, and one in the y-axis after training on the original condition. Interestingly, one of the two participants in the interchanged condition who reached 100 percent accuracy in the z-axis also who found the interchanged condition intuitive. It might be relevant to look at this result in light of the findings of Eitan et al. [90] which indicate that the blind do not share the association between elevation and pitch seen in the sighted, and suggest that the blind may associate pitch with changes in distance. This could also be in line with a finding by Hamilton Fletcher et al. [91] who found that the first mapping presented to some blind participants is perceived by them as the most intuitive.

When asked about how they perceive depth in the day-to-day and whether they experience difficulties, nearly all sighted participants mentioned vision, while some also mentioned the other non-invasive senses, audition, touch, and even smell. On the other hand, the blind and visually impaired, reported a number of distinct experiences with respect to depth perception, among them reference to echoes or other auditory features: "According to the echo coming out of the sound, how close or far it is to you", and "The information I gather is from echoes and listening to the sounds. When I speak, I hear more or less where the wall is next to me because I have an echo coming back to me from it, and then I know more or less how close it is to me". They reported difficulties as follows: "It's not easy for me. . .let's say if I get close to an object, I perceive that there is a figure in front of me. I don't see it but I feel a closed or open space in front of my face." and: "I find myself going in the wrong direction many times. I correct it but I have a limitation with respect to localization in my space." Alongside the use of multiple senses: "Hearing and touch help to find my way." Hearing was particularly mentioned: "I use a cane. Places I don't know, it helps a lot to position myself in the space. Apart from that, I hear the space, by the sounds and noises around you can tell if there is a wall or an object in front of you, sometimes the boundaries of the object. You can identify it by the sounds. When I walk in a corridor, I know when I get close to the wall. By the sound of my steps, for example". These reports emphasize the rationale behind our attempt to design an auditory means of conveying spatial information, that taps into mechanisms that blind and visually impaired individuals are accustomed to and make use of in the day to day.

## Limitations, future directions, and applications

This study is not free from limitations, first of all, the Baseline stage had only 15 trials, resulting in more variability in the baseline results as compared to the test results. Another limitation is that objective exclusion and inclusion criteria regarding hearing abilities (such as audiometry or sound localization tests) were not applied. We also did not inquire into the participants' abilities to discriminate between pitch or their musical background. Possibly the most significant limitation of the study is the small sample size in general, but more specifically of the blind and visually impaired sample population. As such, the findings with respect to these participants in particular cannot be generalized, though they do provide interesting initial

insights, and indicate the usability of the system for people with blindness or visual impairment. In addition, it should be noted that there are differences between early and late blind individuals with respect to spatial abilities and strategies [92–94]. These differences are not explored in this study due to the small sample size of blind participants. Future research could further expand the study within this domain specifically, to fine-tune the algorithm to the needs of the blind (early and late) and visually impaired, and as such explore the system with respect to accessibility, and as a possible tool.

Future applications of the system could integrate the ever-developing breakthroughs from the realm of artificial intelligence, employing image detection for providing content in real-time that adapts to the visual scene. The TLD could be merged with the original algorithm for representing three-dimensional space, using the same representation system and adding back representation of horizontal location using time. Alternatively, a similar approach to the ecologically inspired one could be utilized, for example by changing the balance of the right and left audio channels for indicating horizontal location. In addition, the future directions previously discussed with respect to the algorithm presented in our prior research would be relevant for the current algorithm as well, for example, exploring the expansion of the algorithm to the backward space, or for providing a merged spatial representation of the entire 360 degrees surrounding a person [25, 95]. Another direction would be incorporating tactile feedback, alongside the auditory cues, maximizing the potential for additive effects via multisensory integration.

Regarding the specific development of the algorithm, this research was designed as a proof of concept, and as such the algorithm was built to accentuate differences for achieving an optimal effect. Similar to the algorithm used in previous research, it also relied on a simple manipulation that can potentially be applied at a low (computational and monetary) cost for real world applications. The further refinement of the audio manipulation based on these principles holds significant potential. Building and expanding upon the correlations existing in nature could improve intuitiveness, performance, or both. One such direction of refinement would involve more precise modeling of sound location, specifically incorporating elements of a generalized, or even user specific head-related transfer function [96], or simulating audio effects of specific environments [97]. Such manipulation could potentially dramatically increase performance, especially following training. Another direction could be using an accentuated algorithm to improve performance in specific use cases and ranges, where they would otherwise be difficult to differentiate.

Another particularly promising future direction would be to explore the neural connectivity and activity in the brain corresponding to the use, and training of the system. In past research conducted using the PSVA (described above) sighted blindfolded users were trained on perceiving depth and then their neural activation was explored via positron emission tomography (PET) [50]. The research showed activation in occipito-parietal and occipito-temporal areas [50]. It could be of interest to explore the brain mechanisms correlated with the TLD algorithm via neuroimaging methods. This could be done both during a session of use and before and after a period of training, to test differences related to activation, structure and connectivity in the brain for this type of perception.

A clear understanding of how depth is perceived through the combination of the senses is particularly important today with the rise of extended reality environments that simulate sensory perception. Depth is not necessarily properly conveyed through visual cues alone in VR [98]. In addition, when there is a discrepancy in the sensory signals arriving to the brain from the different senses, it brings about one of the greatest problems facing the widespread adoption of virtual reality, that of motion sickness in virtual reality, also known as cybersickness [99]. As such, this field of research, and specifically the TLD algorithm could provide crucial

insights and practical solutions for accessibility in VR settings, but also possibly increase the immersiveness and embodiment of the user experience.

Finally, the findings of the present study may indicate the potential of perceptual learning with respect to the practical use of sensory substitution devices and extension thereof. For example, this may hint that perceptual learning can be utilized for sensory augmentation aimed at enhancing or extending the senses. These forms of sensory augmentation can stretch the limits of human perceptual abilities, possibly allowing for the application of super senses beyond the capabilities that nature allows for.

## Conclusion

This study demonstrated that the TLD sensory substitution algorithm effectively enables depth and elevation perception through auditory cues in sighted, blind, and visually impaired participants. The findings highlight the intuitiveness of the original mappings, based on auditory depth cues encountered in nature, and the capacity for rapid learning across all participants. Both intrinsic and learned cues provided insight into spatial perception via the algorithm, indicating the potential for novel sensory mappings to be trained in adulthood using SSDs among other methods. The results may suggest that the brain is capable of adapting to new sensory correspondences, intuitions, and capabilities that can be developed beyond traditional critical periods. Finally, this study provides a stepping stone for practical applications in accessibility technologies and virtual reality, offering promising means for enhancing sensory experiences and aiding those with visual impairments. Future studies should further refine the algorithm, explore neural mechanisms by employing neuroimaging, and expand the participant sample size to replicate and generalize these findings.

## Author Contributions

**Conceptualization:** Amber Maimon, Iddo Yehoshua Wald, Amir Amedi.

**Formal analysis:** Meshi Ben Oz.

**Funding acquisition:** Amir Amedi.

**Investigation:** Amber Maimon, Iddo Yehoshua Wald, Meshi Ben Oz.

**Methodology:** Amber Maimon, Iddo Yehoshua Wald, Adi Snir, Meshi Ben Oz.

**Project administration:** Amber Maimon, Iddo Yehoshua Wald.

**Resources:** Amber Maimon, Iddo Yehoshua Wald, Amir Amedi.

**Software:** Adi Snir.

**Supervision:** Amber Maimon, Iddo Yehoshua Wald.

**Visualization:** Amber Maimon, Iddo Yehoshua Wald, Meshi Ben Oz.

**Writing – original draft:** Amber Maimon, Iddo Yehoshua Wald, Meshi Ben Oz.

**Writing – review & editing:** Amber Maimon, Iddo Yehoshua Wald, Amir Amedi.

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
