## [Decision Letter · Decision Letter 0]

6 May 2024

PONE-D-24-09629Perceiving depth beyond sight: evaluating inherent and learned cues via a proof of concept sensory substitution method in the visually impaired and sightedPLOS ONE

Dear Dr. Maimon,

Thank you for submitting your manuscript to PLOS ONE. After careful consideration, we feel that it has merit but does not fully meet PLOS ONE’s publication criteria as it currently stands. Therefore, we invite you to submit a revised version of the manuscript that addresses the points raised during the review process.

Two experts in the field have carefully reviewed the manuscript entitled “Perceiving depth beyond sight: evaluating inherent and learned cues via a proof of concept sensory substitution method in the visually impaired and sighted ”

In light of these reviews and my own reading of the manuscript, I am requesting a major revision and resubmission, in which you will need to respond to each point in each review.
Let me focus on some points that the reviewers and I would like to see addressed. These are:
1)  Important details and background are missing for the auditory parameters used.
2) Overly long writing and several paragraphs with irrelevant information. 
3) The frequent use of the term “ecological” needs clarification, as it does not seem fully justified sometimes.
4) Minor points from me:
Line 111: “Bouba Kiki” is the usual name;
Line 216: “the x and z”: should not be “the y and z”?
Line 220: “and Test - in which participants”...
Line 266 and elsewhere: “Cohen’s d” with capital C
Lines 266 and 267: please revise the punctuation marks     
Line 272: Intechanged                    
There are other points brought out in the reviews and I will carefully attend to your item-by-item responses to them. ==============================

We look forward to receiving your revised manuscript.

Kind regards,

Bruno Alejandro Mesz, Ph.D.

Academic Editor

PLOS ONE

3. We notice that your supplementary figures are included in the manuscript file. Please remove them and upload them with the file type 'Supporting Information'. Please ensure that each Supporting Information file has a legend listed in the manuscript after the references list.

Reviewers' comments:

Reviewer's Responses to Questions

**Comments to the Author**

1. Is the manuscript technically sound, and do the data support the conclusions?

Reviewer #1: Yes

Reviewer #2: Yes

2. Has the statistical analysis been performed appropriately and rigorously? 

Reviewer #1: Yes

Reviewer #2: Yes

3. Have the authors made all data underlying the findings in their manuscript fully available?

Reviewer #1: Yes

Reviewer #2: Yes

4. Is the manuscript presented in an intelligible fashion and written in standard English?

Reviewer #1: Yes

Reviewer #2: Yes

5. Review Comments to the Author

Reviewer #1: The paper reports the behavioral testing of a minimalist sensory substitution method incorporating spoken words that are manipulated using a low-pass filter and pitch to indicate the target locations distance and verticality respectively. This method is compared against an alternative method that swaps the distance and verticality mappings. They find initial divergence in scores (with the initial method being superior) which convergences to equivalent performance after training in the test condition. Overall, the experiment and its findings are fairly straight forward, which should lend itself to a leaner manuscript. Furthermore, important details and background are missing for the auditory parameters used.

The general impression I get is that the writing is frequently overly long which tends to obscure the actual meaning that is attempted to be conveyed. It feels like sentences frequently tend to meander slightly on their way to the point, rather than writing for clarity and efficiency. You could probably eliminate about 30% of the words in this manuscript and have no loss in the actual information provided. By being more exact in the language used, it should make the paper more readable, intuitive, and give yourself space to add crucial information which is missing (e.g. the exact audio used by the system). On top of this, not all writing is relevant enough to be included, so a couple of paragraphs could be cut as well - when evaluating a paragraph ask whether this information directly relates to the experiment, or is this writing for writing's sake?

Finally, there is a bit of overselling at the end about spatial skills being inferred from this experiment, but the experiment itself is an abstract task involving 9 spatially organised buttons. The users are not meaningfully exploring and interacting with an ecologically valid 3D space, so it could be argued that this is closer to an auditory discrimination and memory task, than a distance perception task. This is fine as an experiment, but I think the overall insights that can be gained from this need to be toned down a little as a result.

Below I have comments for each section of the manuscript, I hope the authors find this useful.

Abstract

“ecologically inspired auditory properties” - ecologically tends to mean ‘in the environment’ so would need justification or just that it's a psychological intuition (shown by cross-modal correspondence research). Perhaps a description of the auditory properties is suitable here.

Give actual numbers of sighted, visually-impaired, and blind participants

“Inherentness” is a vague term to be discussed for testing, please be more exact. What effect are you looking for - reaction time, accuracy, participant ratings? It appears to be initial performance results and post-training results. I’m split on whether inherent is the best word choice, perhaps Intrinsic, implicit, initial, baseline, or intuitive might be better choices.

Some of the language is overly long and vague, where more concise specific language would aid readability.

“They also indicate that with training the spatial abilities of the visually impaired and blind can converge with those of the sighted.” Be clear this is talking about the SSD performance.

Introduction

Please add reference for sensory substitution definition.

“The first involves a conversion of full images to audio or tactile” stimulation.

Line 85 onwards - Would be good to explicitly state how spatial information is conveyed in the topospeech approach.

There is a lot of dancing around saying the exact thing that is being done “The implementation proposed above, using ecological representation of depth through auditory properties” when you can just say the auditory properties being used. This would help the reader understand and conceptualise what it is like to experience this sensory stimulation.

“representation of the two axes” which two axes? Are you referring to an inverted correspondence between distance and the proposed ecologically valid auditory property? Or applying these to different axes? In 3D perception, intuitively there are 3 axes, horizontal, vertical, depth. So when two are described, which two?

Looking further into the paper, while it appears distance is being focused on, isn’t verticality being focused on as well? If those are the two axes, they should be explicitly stated. E.g. line 92 “This work expands upon our previous implementation that represented objects and their horizontal and vertical locations, by representing depth.” [and vertical locations]. My understanding at this point is simply that unlike the prior topospeech studies, horizontal localization is the one axis not being explored.

106 - “depth awareness” Depth discrimination as this is probably the most accurate term.

108 - I feel you should be more explicit in the initial sentence that these are psychological intuitions and associations being discussed here.

I find it unusual that word-based synaesthesia is brought up but not abstract forms like music or pitch-related synaesthesias, since you use pitch as one of the axes later on.

122 - “moreare”

The paragraphs on synaesthesia that are not directly related to the experiment could be deleted (lines 125-129). Sticking to visual and audition will help keep the manuscript more concise.

It could be argued that pitch-height correspondences can be considered innate from pre-linguistic infants having a looking preference for bouncing balls that rise and fall with congruent pitch changes (see below). There are a variety of studies for infants, and for whether these associations exist blind and low vision populations.

Walker, P., Bremner, J. G., Mason, U., Spring, J., Mattock, K., Slater, A., & Johnson, S. P. (2010). Preverbal infants’ sensitivity to synaesthetic cross-modality correspondences. Psychological Science, 21(1), 21-25.

Line 139 - you use the acronym SSD but you haven’t introduced it previously.

You could discuss research that shows that SSDs that use cross-modal correspondence have better performance to bolster the idea that you expect this to occur.

If you want to discuss how frequency pass filtering can be used for spatialization (specifically verticality), you could look at ‘virtual acoustic space’ papers or the below reference, which uses band-passed noise to indicate vertical positioning (mimicking auditory changes from the ear pinnae - called head-related transfer function). This may have implications for how the swapped condition performs with low-pass filtering is used for verticality.

González-Mora, J. L., Rodriguez-Hernandez, A., Burunat, E., Martin, F., & Castellano, M. A. (2006, April). Seeing the world by hearing: Virtual Acoustic Space (VAS) a new space perception system for blind people. In 2006 2nd International Conference on Information & Communication Technologies (Vol. 1, pp. 837-842). IEEE.

Richardson, M., Thar, J., Alvarez, J., Borchers, J., Ward, J., & Hamilton-Fletcher, G. (2019). How much spatial information is lost in the sensory substitution process? Comparing visual, tactile, and auditory approaches. Perception, 48(11), 1079-1103.

Line 141-143 - or no preference would yield the same result

I feel like the introduction overall could be pruned and focused more by asking ‘is this point directly relevant to the study, or is this a tangent?’ and eliminate all the tangents. This would give space for other more relevant information and arguments, such as on pitch-height and pitch-distance associations. Cross-modal correspondences improving user performance on SSDs. Effect of blindness on cross-modal correspondences. Or exploring the auditory characteristics of distance perception (i.e. “the ecological cues”).

Line 172 - I thought the prior algorithm did not do depth? I presume the algorithm altered or repurposed for this study?

Ecological is used to refer to the distance parameters and scene richness. But also the experiment is abstracting out a scene and not providing rich visual detail, so this is more of a discussion point than a goal of the present research. No evaluation of verticality is mentioned here. I feel like lines 171-181 could be more ‘to-the-point’.

Methods

- “Out of the participants, five were blind, and two were visually impaired.” from the information provided the age and gender of the blind and visually-impaired participants cannot be determined.

“Recruitment for the study took place between May 14, 2023 to December 13, 2023” don’t think this information is needed.

“it uses a sweep line technique to scan a visual scene from left to right. It was utilized for conveying spatial information in the x and y axes” If the scanline is not altering the pitches used, then its just utilized for the X axis here.

“The y-axis location was represented by pitch, as in the previous algorithm” which pitches?

I take it there is no scan line in this new algorithm version?

The writing is overly long and misses some methodological detail that should be included.

I would like the technical specifics on how the auditory properties of distance were made. What distances are being simulated by the subtractive synthesis (e.g. just within arm reaching distance, or did you simulate much further distances to exaggerate the changes in audio). Ecologically valid is perhaps overselling it unless you are replicating actual distances very precisely. To me it's more ecologically inspired mappings. How were the words made, are these synthetized voices or verbal recordings, we should not need to look up the other paper to know.

The baseline stage should be a separate paragraph since this is an important comparison made in the paper.

Baseline only has 15 trials while the test has 90 trials. This should be mentioned as a limitation as there could be way more variability in the baseline results.

Results

Try to be consistent in the naming of the conditions - e.g. Figure 2 is called the ‘reversed condition’ rather than interchanged. I think swapped is a better descriptor than interchanged or reversed.

Figure 4 should have a visual key on the meaning of different line colors and dotted lines.

If pitch appears to have a strong influence on whether the subject is correct or not, its important to note which pitches are used to help explain why that might be the case. Also subjects ability to discriminate pitch or their musical background would help elucidate these findings.

Figure 5 - there is a lot of dead space in these diagrams, which could be filled with explicit mentions of what audio is being heard in these - e.g. “High pitch, Low synthesis”

Ultimately I still don’t know the pitches or exact synthesis (i.e. what frequencies are being band pass filtered) are used in any of these conditions. If the vocal pitch is being altered, do low and high vocals sound like different genders? A spectrogram of these different audio mixes would be good to see, for example, the same word, altered in pitch and subtractive synthesis.

So far it appears that the audio used, rather than what it represented is the key factor determining their success rates (after training). This might be expected if the key influencing factor is auditory discrimination rather than what it represents visually/spatially.

Figure 6 should have a title for each graph. Reminders on what each Z and Y axis represents (pitch, synthesis) would be appreciated are important when things are constantly switching back and forth. You could have patterning/outlines indicate whether the subject is hearing pitch changes or synthesis changes.

Please illustrate significant comparisons on the graphs themselves. Please use colors to help separate the regular and reversed conditions so it is clear what everything related to visually. E.g. all regular data could be various shades of blue, while all reversed could be various shades of orange. Baseline can be lighter shades, and test can be darker shades (like you are currently doing). It’s probably better to have the condition (regular, reversed) stated first and the time (baseline, test) stated second on the graphs.

Everything so far has been baseline then test, and now in Table 3 its reversed? Please keep baseline than test consistent.

Figure 7 - “axix” in title. If the diagrams are flipped, for the 2 diagrams on the right, shouldn’t the axes read columns (Y-axis) and rows (X-axis)? Because don’t the columns and rows refer to the stimuli subjects are interacting with and you’ve just ‘flipped’ the diagram, which means you should have also flipped the axes? Color axis is not named.

Discussion

How will X, Y, and Z be combined?

“Ecological principles” is still a very vague way of describing a band-pass filter.

This study explores depth perception, but also verticality, there is just as much evidence on verticality as there is depth.

In order to discuss when sensory substitution is intuitive, reference to the following paper would help a lot in knowing what has been explored previously

Stiles, N. R., & Shimojo, S. (2015). Auditory sensory substitution is intuitive and automatic with texture stimuli. Scientific reports, 5(1), 15628.

“indicating a high effectiveness for the training, and a high general learnability for these correspondences.” Are we considering the reversed condition also correspondences?

If pitch works well with both elevation and proximity, there are a few papers on this as well as the following reference by Eitan:

Eitan, Z., Ornoy, E., & Granot, R. Y. (2012). Listening in the dark: Congenital and early blindness and cross-domain mappings in music. Psychomusicology: Music, Mind, and Brain, 22(1), 33.

“steep learning curve with training times averaging under 15 minutes overall” Steep learning curve can also be interpreted as ‘very difficult to learn’, which is not the case here, so alternative phrasing may help clarity.

“The significant differences in success rates in the original condition compared to the

interchanged one at baseline indicate that the chosen mapping was effective in building upon existing crossmodal correspondences and ecological cues.” This sentence is a good example of a general tendency throughout the writing, to use more words than required, when fewer can be easier to read and clearer - e.g. “At baseline, the original mapping significantly outperformed the reverse mapping, indicating the utility of crossmodal correspondences and ecological cues.”

“It is interesting to note that two blind participants were the only two to report that they

perceived the interchanged condition to be intuitive.”

This is reminiscent of a finding in this abstract that the first presented mapping is the most intuitive for some blind subjects:

Hamilton-Fletcher, G., Pieniak, M., Stefanczyk, M., Chan, K., & Oleszkiewicz, A. (2020). Visual Experience influences associations between Pitch and Distance, but not Pitch and Height. Journal of Vision, 20(11), 1316-1316.

Not all paragraphs are relevant to the specific research being conducted here, and instead feel like general writing about a topic that is unnecessary. For example lines 519-527. This is about how vision is stronger than audition for 3D discrimination, but so what? What relevance does this have for your specific questions, or sensory substitution in general? How does it influence SSD design?

There is a big difference between congenitally blind and late blind that is not commented on here, where it has an impact on their existing spatial abilities.

“Overall, it has been suggested that blind individuals have inferior spatial capabilities or perceptual abilities than the sighted.” This is way too broad a statement when the reference is about inferring visual perspective cues from sensory substitution. Monica Gori’s work on how visual impairments alter the perception of distance is far more suitable for a paragraph on general spatial hearing skills.

Line 537 - Some of the language here is way too total for what is sometimes a minor difference in spatial discrimination skills between sighted and blind (early/late) individuals.

Don’t start a paragraph with ‘and yet’

“This is unlike other visual-to-auditory sensory substitution devices that use arbitrary mappings of visual information to sounds.” Are they arbitrary? Are you considering cross-modal correspondences arbitrary? Most SSDs tend to use some cross-modal correspondences.

The subject quotes sections could be heavily reduced, just to pick out the super relevant points.

Reviewer #2: General assessment

This is an interesting manuscript that explored the TopoLanguageDepth sensory substitution method for spatial perception, particularly depth perception, via a combination of auditory cues: naming objects and representing their elevation and depth. The authors explored the role of inherent and learned cues in depth perception as well as training of sensory mappings by perceptual learning through audition.

It is a proof of concept that included participants with normal vision (n=33), visually impaired (n=2), and blind (n=5) participants.

It is a well written manuscript with minor typos.However, I have some suggestions to improve the manuscript.

Abstract

There is a typo in “...by comparing it to an in interchanged…”

Introduction

There are repetitive mentions of the goals of the study (Ln95, 99, 105, 171). I suggest integrating them only in the final paragraph of the Introduction.

Ln139 Please, clarify the acronym “SSD” in its first mention

Ln153-161, I suggest to improve the transition from the previous sentences, related to sight, and this section focused on audition.

Methods

Ln184 Please include units in descriptive statistics (ie. years/yr)

Ln 192 Please, clarify the Participants section, particularly how and where they were recruited. Were the participants patients or students from Reichman University?

Did you perform a sample size calculation?

What were the inclusion/exclusion criteria?

Ln193 I suggest to include the experiment conditions section, and clarify the groups (“conditions”; original and interchanged?) as well as how the visually impaired and blind participants were allocated to those groups.

Ln196-202 I suggest that this information be removed or included in the Introduction so that this section is focused on the modifications from the original algorithm

Ln206 Please justify naming “z-axis” when applying cues intended for depth perception when the representation is still in a 2D board. I suggest y1-axis and y2-axis to differentiate between conditions

Ln260-262 There is no further mention of this interview, its structure, domains as well as the 9 questions. What was the purpose of this interview and where are the results? Please, clarify

Ln262 The authors should include a Statistical Analysis section, identifying the types of variables and their measurements, data processing, and the rationale for the statistical tests performed.

Results

For all figures and tables, plese verify and homologate the nomenclature. Also, include more informative captions: units, groups, auditory cues, statistical tests performed,...

Table 1

I recommend to include 95% confidence intervals when reporting the Means for a better interpretation of the uncertainty of the estimate

Table 2

I suggesto to combine Table 2 and 4

Figure 2:

Are regular and reversed conditions referring to original and interchanged conditions? Please clarify and maintain a consistent nomenclature

Please differentiate visually impaired and blind subjects in the Subjects column. That would help to identify where the visually impaired (n=2) and blind (n=5) participants were grouped, their success rate, individual and compared to the overall average.

Figure 3.

Please clarify “Reversed” and “Regular”

Figure 4

I suggest that more contrasting colors be applied to differentiate between blind/visually impaired from sighted participants

Figure 5

Please include the rationale for the flipped diagram in the Methods - Data analysis or Statistical analysis section

It is not clear to me why was this performed, and why only for the interchanged condition instead of both conditions. In this regard, and if the figure does not changes, I suggest to position the flipped diagram below its corresponding panel (interchanged).

How inherent and learned cues participate in depth perception?

Please, clarify how spatial abilities of visually impaired and blind can converge with those of the sighted?

Please verify that the supplementary material be mentioned in the manuscript.

Ln326 and 360 Please clarify substractive synthesis for elevation instead of distance (Ln206)

Discussion

Ln352 Along the manuscript, there is a vast diversity of mentions related to the “ecological” domain such as “ecological principles”, “ecological mapping”, “ecological information”, “ecological behavior”, “ecological considerations”, “ecological cues”, “ecologically inspired”, “ecological representation” of auditory properties. Please clarify that domain since its first mention in the Introduction section (what is meant by "ecological" in this context?), and discuss the related results accordingly (ie. how well the algorithm captures and represents auditory cues in a manner consistent with natural depth or distance perception?).

Ln564 Were the one or two participants that reached 100% accuracy in the y-axis or z-axis, respectively, the participants that found intuitive the interchanged condition?

Ln597 I suggest to highlight the small sample size, in general, and particularly for blind and visually impaired participants. In addition, as this was a proof of concept and a sample size calculation was not performed, the findings cannot be generalized. Additional limitations might include the lack of inclusion/exclusion criteria that might impact your results such as hearing functions (ie. audiometry or sound localization tests)

Conclusions

Please, include this section

6. PLOS authors have the option to publish the peer review history of their article (what does this mean?). If published, this will include your full peer review and any attached files.

Reviewer #1: No

Reviewer #2: **Yes: **José Darío Martínez-Ezquerro

---

## [Author Response · Author response to Decision Letter 0]

21 Jun 2024

Thank you for giving us the opportunity to revise the manuscript. We have responded to the reviewers' comments at length in the attached document titled "Replies to reviewers".

---

## [Decision Letter · Decision Letter 1]

23 Aug 2024

Perceiving depth beyond sight: evaluating intrinsic and learned cues via a proof of concept sensory substitution method in the visually impaired and sighted

PONE-D-24-09629R1

Dear Dr. Maimon,

We’re pleased to inform you that your manuscript has been judged scientifically suitable for publication and will be formally accepted for publication once it meets all outstanding technical requirements.

Kind regards,

Bruno Alejandro Mesz, Ph.D.

Academic Editor

PLOS ONE

Additional Editor Comments: Dear authors, congratulations on the acceptance of your work!. Please note that according to Plos ONE policy, **it is essential** that you provide a stable link to the data and a Data Availability Statement in the manuscript:

"The PLOS Data policy requires authors to make all data underlying the findings described in their manuscript fully available without restriction, with rare exception (please refer to the Data Availability Statement in the manuscript PDF file). The data should be provided as part of the manuscript or its supporting information, or deposited to a public repository. For example, in addition to summary statistics, the data points behind means, medians and variance measures should be available. If there are restrictions on publicly sharing data—e.g. participant privacy or use of data from a third party—those must be specified".

"A Data Availability Statement describing where the data can be found is required at submission. Your answers to this question constitute the Data Availability Statement and will be published in the article, if accepted". 

Reviewers' comments:

Reviewer's Responses to Questions

**Comments to the Author**

1. If the authors have adequately addressed your comments raised in a previous round of review and you feel that this manuscript is now acceptable for publication, you may indicate that here to bypass the “Comments to the Author” section, enter your conflict of interest statement in the “Confidential to Editor” section, and submit your "Accept" recommendation.

Reviewer #2: All comments have been addressed

Reviewer #3: All comments have been addressed

2. Is the manuscript technically sound, and do the data support the conclusions?

Reviewer #2: Yes

Reviewer #3: Yes

3. Has the statistical analysis been performed appropriately and rigorously? 

Reviewer #2: Yes

Reviewer #3: Yes

4. Have the authors made all data underlying the findings in their manuscript fully available?

Reviewer #2: Yes

Reviewer #3: No

5. Is the manuscript presented in an intelligible fashion and written in standard English?

Reviewer #2: Yes

Reviewer #3: Yes

6. Review Comments to the Author

Reviewer #2: I am pleased to see the substantial revisions made to this manuscript, which have significantly enhanced its clarity and rigor. The paper now presents a well-structured and compelling contribution to the field of depth perception.

This article expands the exploration of depth perception beyond the traditional focus on vision, incorporating audition as a key sensory modality. Although a proof-of-concept, the evidence provided supports the notion that certain cross-modal correspondences may not only be innate but also learnable. The findings highlight the potential of sensory substitution devices and algorithms to facilitate novel sensory-perceptual mappings, demonstrating that individuals can be effectively trained to develop new sensory associations.

I have only a minor suggestion that the authors may consider if they find it appropriate:

In Ln 137, I suggest modifying the phrase: “A third novel cross-modal ecological sensory substitution algorithm implemented in this study…”. This change could help future literature reviews more effectively summarize the central approaches for representing spatial information via audio or touch, and it might also support further development of research based on your approach for depth perception.

Congratulations to the authors on a well-executed study. I look forward to seeing this important work published.

Reviewer #3: The manuscript presents a spatial localisation task through auditory sensory subsitution perception, with sighted, blind, and visually impaired participants. I'm coming to this review process after the manuscript has already been revised and can see that the previous reviewers' comments were insightful and contributed to a much improved manuscript. I feel that the revised version has suitibly addressed their comments and I have no further comments of my own.

The only thing I noticed was that the authors have stated that the data is freely available, but did not change the XXX in the text to provide a link ('Describe where the data may be found in full sentences. If you are copying our sample text, replace any instances of XXX with the appropriate details.'), nor could I find a data availability section in the manuscript. Please update this, in line with Plos One policy, to provide a long term stable link to a data respository.

7. PLOS authors have the option to publish the peer review history of their article (what does this mean?). If published, this will include your full peer review and any attached files.

Reviewer #2: **Yes: **José Darío Martínez-Ezquerro

Reviewer #3: **Yes: **Mike Richardson

---

## [Editor Report · Acceptance letter]

16 Sep 2024

PONE-D-24-09629R1 

PLOS ONE

Dear Dr. Maimon, 

I'm pleased to inform you that your manuscript has been deemed suitable for publication in PLOS ONE. Congratulations! Your manuscript is now being handed over to our production team.

Kind regards, 

on behalf of

Dr. Bruno Alejandro Mesz 

Academic Editor

PLOS ONE